# Supercritical Fluid Extraction of Oils from Cactus *Opuntia ficus-indica* L. and *Opuntia dillenii* Seeds

**DOI:** 10.3390/foods12030618

**Published:** 2023-02-01

**Authors:** Ghanya Al-Naqeb, Cinzia Cafarella, Eugenio Aprea, Giovanna Ferrentino, Alessandra Gasparini, Chiara Buzzanca, Giuseppe Micalizzi, Paola Dugo, Luigi Mondello, Francesca Rigano

**Affiliations:** 1Center Agriculture Food Environment (C3A), University of Trento, 38098 Trento, Italy; 2Department of Food Sciences and Nutrition, Faculty of Agriculture Food and Environment, University of Sana’a, Sana’a P.O. Box 1247, Yemen; 3Department of Chemical, Biological, Pharmaceutical and Environmental Sciences, University of Messina, 98168 Messina, Italy; 4Faculty of Science and Technology, Free University of Bozen-Bolzano, Piazza Università 5, 39100 Bolzano, Italy; 5Chromaleont s.r.l., c/o, Department of Chemical, Biological, Pharmaceutical and Environmental Sciences, University of Messina, 98168 Messina, Italy; 6Unit of Food Science and Nutrition, Department of Medicine, University Campus Bio-Medico of Rome, 00128 Rome, Italy

**Keywords:** *Opuntia ficus-indica* L., *Opuntia dillenii*, supercritical fluid extraction, antioxidant activity, chemical characterization

## Abstract

This study aimed to assess the capability of supercritical fluid extraction (SFE) as an alternative and green technique compared to Soxhlet extraction for the production of oils from *Opuntia ficus-indica* (OFI) seeds originating from Yemen and Italy and *Opuntia dillenii* (OD) seeds from Yemen. The following parameters were used for SFE extraction: a pressure of 300 bar, a CO_2_ flow rate of 1 L/h, and temperatures of 40 and 60 °C. The chemical composition, including the fatty acids and tocopherols (vitamin E) of the oils, was determined using chromatographic methods. The highest yield was achieved with Soxhlet extraction. The oils obtained with the different extraction procedures were all characterized by a high level of unsaturated fatty acids. Linoleic acid (≤62% in all samples) was the most abundant one, followed by oleic and vaccenic acid. Thirty triacylglycerols (TAGs) were identified in both OFI and OD seed oils, with trilinolein being the most abundant (29–35%). Vanillin, 4-hydroxybenzaldehyde, vanillic acid, and hydroxytyrosol were phenols detected in both OFI and OD oils. The highest γ-tocopherol content (177 ± 0.23 mg/100 g) was obtained through the SFE of OFI seeds from Yemen. Overall, the results highlighted the potential of SFE as green technology to obtain oils suitable for functional food and nutraceutical products.

## 1. Introduction

*Opuntia* is a genus of plants belonging to the Cactaceae family, the latter counting around 1800 species. Among them, approximately 1400 species belong to the genus *Opuntia* and are distributed in Mediterranean countries, Mexico, Europe, and other areas [1]. *Opuntia ficus-indica* L. (OFI) and *Opuntia dillenii* (OD) are the most frequently used species for human consumption. In the traditional medicine of many countries, they are used to treat some diseases and disorders, such as whooping cough and eye inflammation, or as anti-ulcerogenic or antidiarrheal agents [2]. OFI, commonly called prickly pear, is cultivated in many areas of Yemen, where it is used as food, feed, and also as prepared products such as juice, jam, and cosmetics, thus making this plant extremely advantageous from an economic point of view [3,4]. The fruits of OFI contain about 9–10% of seeds [5], which are reported to contain 5–16% of oil [3,6]. Various researchers have reported the fatty acids composition of OFI oil, which is rich in unsaturated fatty acids (80–88%) such as linoleic acid (49.3–78.8%), oleic acid (12.8–25.3%), vaccenic acid (4.3–6.3%), and linolenic acid (0.23–1.1%). The main saturated fatty acids are palmitic (9.3–14.3%) and stearic (2.2–4.3%) acids [7,8,9,10]. Seven different phenolic compounds have been identified in the OFI oil. They belong to three families: hydroxyl cinnamic acid derivates (p-coumaric acid, p-coumaric acid ethyl ester, ferulic acid), hydroxyl cinnamaldehyde derivates (furaldehyde), and hydroxybenzaldehyde derivates (4-hydroxy benzaldehyde, vanillin, syringaldehyde) with vanillin, syringaldehyde, and ferulaldehyde detected at the highest level [11]. The total amount of tocopherols in OFI oil was reported in the range between 3.9 and 50.0 mg/100 g [12], where γ-tocopherol showed 94.12% of the total vitamin E content [13]. The triacylglycerols (TAGs) profile of OFI oil was reported, with trilinolein and oleyldilinolein being the main ones, with an average percentage of 25.6% and 21.5%, respectively [13].

Concerning OD, it is a wild plant that can be found in different places in Yemen, especially in Taiz and Hodeida. The OD fruits have a sour taste and big seeds (Figure 1). Some people in Yemen make fresh and concentrated juice from the OD fruits, thus discarding a large number of seeds. OD seeds have been reported to contain 6.65–13.12% of oil [7,8,9], which are rich in bioactive molecules, such as mono- and polyunsaturated fatty acids (MUFAs and PUFAs) and vitamin E. At the moment, only a limited number of studies deal with a comprehensive chemical characterization of the oil. To the best of the authors’ knowledge, the only work reporting a quite detailed investigation of the chemical composition of OD oil was carried out on a Moroccan species after a conventional solvent extraction procedure [8].

On the other hand, an impressive number of studies have recently shown that the seed oils of both OFI and OD possess good nutritional value and multiple health benefits, including antioxidant activity both in vivo [10] and in vitro [8,11], antimicrobial [12], anti-diabetic [3,13,14], lipid-lowering effect [15], in vitro anticancer effect [16], anti-inflammatory, and antiulcer effects [10,11].

Due to the multiple health benefits of both OFI and OD seed oils, an efficient extraction process is of prime importance to preserve the quality of these oils. Different extraction processes have been applied for OFI and OD seeds, including maceration [10], cold press [14,17,18], ultrasound-assisted extraction [19], and Soxhlet [20,21].

However, such methods are time-consuming, have environmental hazards, and have negative implications for operator safety. Moreover, they might have a threat to consumer health if the organic solvents are not completely removed. Recently, supercritical fluid extraction (SFE) has appeared to be a valid alternative for seed oil extraction and has received considerable attention. The SFE system has many advantages when carbon dioxide (CO_2_) is used as a solvent, which is not toxic, environmentally friendly, non-explosive, and of food grade [17,22,23]. Furthermore, the oil can be extracted under low temperatures and oxygen-free conditions [20]. Studies have shown that SFE-extracted oils are enriched with antioxidant compounds [21,22].

There is quite a poor report of the literature data on the SFE extraction of both OFI and OD seeds with detailed chemical characterization, especially triacylglycerol profiles (TAGs), vitamin E, and phenolic compounds, have been found. Rather, only the total fatty acid (FA) composition was elucidated, and the antioxidant activity was assessed [9,23].

In the present study, oils from OFI seeds originating from Yemen (YV1) and Italy (I/S) and OD (YV2) originating from Yemen were extracted using the SFE system and compared to the one obtained from a Soxhlet extraction. The recovered oils were comprehensively characterized in terms of their chemical composition, including fatty acids, tocopherols (vitamin E), and polyphenols.

## 2. Materials and Methods

### 2.1. Chemicals

Analytical standards of trinonanoin, triundecanoin, tritridecanoin, tripentadecanoin, triheptadecanoin, trinonadecanoin, and the C4 -C24 even carbon saturated FAMEs standard mixture (1000 µg/mL each in n-hexane), n-heptane, methanol, acetic acid (reagent grade), potassium hydroxide (KOH), acetonitrile, methanol, 2-propanol (LC-MS grade), n-hexane (HPLC grade) were purchased from Merck Life Science (Darmstadt, Germany).

### 2.2. Samples Preparation

Three varieties of prickly pear fruits were used in this study: OFI from Yemen (YV1), OD (YV2) from Yemen, and OFI from Italy/Sicily (I/S). Seeds were separated from the fruits, naturally dried at room temperature, packed in plastic bags, and stored at 4 °C until used. Dried seeds were milled using a hammer miller (Mill- LM3100, Perten Instruments, Sweden). The resulting seeds powder showed a particle size distribution of a diameter between 250 and 100 μm, determined by an orbital sieve shaker (Retsch GmbH, Verder Scientific, Haan, Germany).

### 2.3. Supercritical Fluid Extraction

SFE was performed in a semi-batch pilot system (Superfluidi s.r.l., Padova, Italy) as previously described [22] with some modifications. For each extraction run, the basket was filled with about 100 g of milled seeds. The following processing parameters were used: a pressure of 300 bar, a CO_2_ flow rate of 1 L/h, and temperatures of 40 °C and 60 °C. The selection of the operative parameters was based on previous experience with SFE performed on other seeds [23,24]. The pressure was set at the highest level of the system. Two temperatures were tested to assess their effect on the yield. During each extraction, the oil was collected every 10 min from the separator while the CO_2_ was released. The extraction time was equal to 2.30 h.

### 2.4. Soxhlet Extraction

For the Soxhlet extraction, n-hexane was used in order to recover oils from the powdered seeds. Batches of 20 g either of milled OFI or OD seeds were placed in each extraction thimble, and 150 mL of n-hexane was used. The extraction was carried out for 4 h. Then, the solvent was evaporated from the oil using a rotary evaporator (LABOROTA 4000, Heidolph, Schwabach, Germany). The oil yield was calculated on a dry weight basis as a ratio between the amount of oil and the number of ground seeds used for the extraction:(1)$$\mathrm{Yield}\;\left(\%\right)=100\times \frac{\mathrm{Weight}\;\mathrm{of}\;\mathrm{extracted}\;\mathrm{oil}}{\;\mathrm{Weight}\;\mathrm{of}\;\mathrm{initial}\;\mathrm{used}\;\mathrm{seeds}}$$

### 2.5. Chemical Characterization of OFI and OD Seeds Oil

#### 2.5.1. Determination of the Total Fatty Acid Composition by GC Analyses

Fatty acid methyl esters (FAMEs) were obtained by a cold transesterification reaction as previously reported by Ciriminna et al. [24]. Briefly, 50 mg of oil was dissolved in 1 mL of n-heptane and added to 0.1 mL of a 2N solution of potassium hydroxide (KOH) in methanol. The mixture was stirred for 5 min at room temperature. The heptane upper layer was transferred into a 1.5 mL autosampler vial prior to GC-MS and GC-FID (FID, flame ionization detector) analyses for FAME identification and quantification, respectively.

Specifically, GC-MS analyses were performed using a GC-2010 (Shimadzu, Duisburg, Germany) gas chromatograph coupled to a single quadrupole mass spectrometer (QP2020, Shimadzu). An AOC-20i autosampler and split/splitless injector were installed on the GC system. The separation of the analytes was carried out by using an SLB-IL60 (ionic liquid, IL) 30 m × 0.25 mm ID × 0.20 µm df (Merck Life Science) capillary column, operated under the following temperature program: from 50 °C to 280 °C at 3 °C min^−1^. Helium was utilized as a carrier gas at a constant linear velocity of 30 cm s^−1^ and initial inlet pressure of 26.6 kPa. The injection volume, injector temperature, and split ratio were 0.2 µL, 280 °C, and 1:100, respectively. A single quadrupole MS detector was operated in the scan mode (mass range of 40–550 *m*/*z*); the interface and ion source temperatures were 250 °C and 220 °C, respectively. The GC-MS solution software (version 4.50 Shimadzu) was used for data collection and handling. A dedicated mass spectra database, namely LIPIDS ver. 1.0 (Shimadzu) was used for the spectral match. Moreover, a lab-constructed linear retention index (LRI) database [25] was employed for LRI matching, according to an automatic data processing method, by applying a dual-filter identification (minimum MS spectral similarity of 85% and LRI tolerance of ±10 units). In such respect, the C4-C24 FAMEs standard solution was injected prior to the analysis under the same experimental conditions for the calculation of LRIs for the detected peaks.

GC-FID analyses were carried out using a GC-2010 instrument (Shimadzu) equipped with a flame ionization detector (FID). The GC column, temperature program, carrier gas, linear velocity, and injection conditions were the same as described for the GC-MS system. The initial inlet pressure was 99.4 kPa. The FID temperature was set at 280 °C. The following FID gas flows were set: 40 mL min^−1^ for H_2_, 30 mL min^−1^ for the make-up gas (N_2_), and 400 mL min^−1^ for air. Data were collected and processed using the LabSolution software (version 5.92, Shimadzu, Duisburg, Germany). All samples were analyzed in triplicate.

#### 2.5.2. Determination of the Triacylglycerol Composition by LC/MS Analysis

An amount of 10 mg of triundecanoin (C11C11C11) was added as an internal standard (IS) to 20 mg of oil, and the mixture was dissolved in 2-propanol, up to a final volume of 1 mL, prior to the LC-MS analysis. The instrumental setup consisted of a Nexera UHPLC system coupled to an LCMS-2020 spectrometer through an atmospheric pressure chemical ionization (APCI) source (Shimadzu, Duisburg, Germany). Separations were performed on an Ascentis Express C18 10 cm × 2.1 mm, 2.7 µ m d.p. columns (Merck Life Science, Darmstadt, Germany). The employed mobile phases were acetonitrile (A) and 2-propanol (B), and the linear gradient program was as follows: 0–52 min, 0–70% B, held for 3 min. The flow rate was 0.5 mL min^−1^, the oven temperature was 35 °C, and the injection volume was 5 µL.

The following MS parameters, through the APCI source in a positive (+) ionization mode, were employed: the interface, desolvation line, and heat block temperatures were set at 450 °C, 250 °C, and 300 °C, respectively; nebulizing gas and drying gas flow (N_2_) were set at 1.5 L min^−1^ and 5 L min^−1^, respectively; and the acquisition MS range was 250–1200 *m*/*z* with an event time of 1 s. Data were acquired by using LabSolution ver. 5.95 software (Shimadzu, Duisburg, Germany) and processed through the recently implemented ChromLinear software.

For qualitative purposes, a reference standard mixture of odd chain carbon number triacylglycerols (TAGs) from trinonanoin (C9C9C9) to trinonadecanoin (C19C19C19) was injected at the beginning and at the end of the analytical batch in order to automatically calculate LRI for all the peaks of the sample. The ChromLinear software was able to match the calculated LRI and acquired MS spectra for single TAGs with the previously built LRI database and MS spectral library according to an automatic and rapid dual-filter identification strategy, similar to GC-MS analyses. For quantitative purposes, the normalized peak areas (ratio between the analyte and IS peak areas) were used for relative quantification, considering a quite similar MS response for all the identified TAGs.

#### 2.5.3. LC-FLD Analysis of Vitamin E

Vitamin E determination was carried out as previously described by Dugo et al. [26] with a slight modification. Briefly, 1 mL of oil was dissolved in n-hexane up to a final volume of 10 mL prior to LC-FLD analysis. The employed analytical system consisted of a Nexera-X2 system (Shimadzu, Milan, Italy) equipped with an RF-20AXS fluorescence detector, as described in previous work. Data acquisition and processing were performed by the LCMS solution Ver. 5.85 software (Shimadzu, Duisburg, Germany). The chromatographic separation was carried out on an Ascentis Si column, 250 × 4.6 mm L × I.D., with a particle size of 5 μm (Merck KGaA, Darmstadt, Germany), kept at 25 °C and operated in isocratic elution mode (n-hexane/isopropanol 99:1, *v*/*v*) at a flow rate of 1.7 mL min^−1^. The injection volume was 5 μL. The peaks were recorded using 290 nm as the excitation wavelength and 330 nm as the emission wavelength. A previously validated method by Dugo et al. [26], based on external calibration, was used for the quali-quantitative determination of the detected vitamers.

#### 2.5.4. LC Analyses of Phenolic Compounds

Phenolic compounds were recovered from the oil by liquid–liquid extraction, using methanol: water 80:20 (*v*:*v*) mixture, according to a 1:1 ratio sample: solvent (*w*:*v*), after the dissolution of the oil in n-hexane. Briefly, 0.5 g of oil was dissolved in 0.5 mL of n-hexane and added to 0.5 mL of methanol: water 80:20 (*v*:*v*). The mixture was sonicated for 5 min and centrifuged for 10 min at 3000 rpm. The remaining n-hexane phase was extracted two more times under the same conditions, and the three methanolic aqueous phases were pooled, washed with 0.5 mL of n-hexane, and concentrated to dryness under a nitrogen stream. The residue product was reconstituted in 200 µL of methanol prior to injection into the LC instrument, which was equipped with a photodiode array detector (PDA) SPD-M20A directly connected to the LC column outlet and serially coupled to an LCMS-2020 spectrometer via an electrospray (ESI) source for mass spectrometry (MS). The chromatographic separation was achieved on an Ascentis Express RP C18 (150 mm × 4.6 mm ID × 2.7 µm) column, operated at a temperature of 35 °C and at a flow rate of 1.0 mL min^−1^. The mobile phase consisted of (A) water with 0.15% of acetic acid and (B) acetonitrile, according to the following gradient program: 0–30 min, 2–30% B, 30–40 min, 30–65%, held for 10 min; the injection volume was 5 µL. PDA detection was applied in the range of 200–700 nm with a sampling frequency of 4.1667 Hz and a time constant of 0.480 s. Chromatograms were extracted at 280 nm. ESI-MS acquisition was performed in the mass range 100–800 *m*/*z* with an event time of 1 s; nebulizing and drying gas (N_2_) was set at 1.5 and 5.0 L/min, respectively. DL and heat block temperatures were set at 300 and 350 °C, respectively.

### 2.6. Antioxidant Activity by DPPH Assay

The antioxidant activity of the OFI and OD seed oils extracted by SFE and Soxhlet was evaluated using the 1,1-diphenyl-2-picrylhydrazil assay (DPPH) as described by Brand-Williams et al. [27]. The extracted oils (250 mg) from YV1, YV2, and I/S seeds were mixed with 1.5 mL of methanol and sonicated for 5 min at 25 °C three times. The methanolic extract was collected and used to perform the analysis. The DPPH reagent was prepared by dissolving 10 mg in 250 mL of methanol. The measurements were performed by transferring about 1.9 mL of DPPH solution into the cuvettes and adding 100 μL of the methanolic extracts. The samples were stored in the dark for 1 h at room temperature. The absorbance was measured at 515 nm with a spectrophotometer (Cary 100 Series UV–Vis Spectrophotometer, Agilent Technologies, Milano, Italy). The antioxidant activity of the oils was determined using a standard calibration curve based on Trolox (6-hidroxy-2,5,7,8-tetramethylchroman-2-carboxylic acid, from Sigma–Aldrich Co, St. Louis, EUA) as a reference antioxidant. The analysis was performed in triplicate, and the results were expressed as the Trolox equivalent value per gram of oil (mg Trolox/g oil).

### 2.7. Statistical Analysis

All parameters obtained for fatty acids, TAGs, and phenolic compounds were statistically analyzed to detect significant differences (α = 0.001) among the samples by a one-way ANOVA performed through the XLSTAT software. The post-hoc HSD Tukey test, when appropriate, was applied to find out which specific groups’ means (compared with each other) were different. Data obtained from oil yield and antioxidant activity were analyzed using one-way ANOVA GraphPad Prism 7. The number of independent experiments, details on statistical comparisons, and levels of significance were indicated in the captions of the respective figures and tables.

## 3. Results and Discussion

### 3.1. Oil Yields

The yield of the seed oils from the three different samples of OFI and OD seeds was highly dependent on the extraction method, the temperature applied during SFE, and the varieties. The results showed that the highest yields were obtained through Soxhlet extraction with 12 ± 1.35% for YV1, 10 ± 0.89% for YV2, and 11.8 ± 1.34% for I/S compared to 6.2 ± 1.27, 8.3 ± 1.34% for YV1, 5.7 ± 0.96, 7.4 ± 1.21% for YV2 and 6.3 ± 0.76, and 7.7 ± 1.31% for I/S by SFE extraction at 40 °C and 60 °C, respectively.

For the SFE method at a fixed pressure and different temperatures, the extraction yields increased from 6.2 ± 1.27% to 8.3 ± 1.34% in YV1, changing the temperature from 40 °C to 60 °C. Basically, the YV1 sample originated from Yemen was characterized by the highest yield when extracted by Soxhlet and the SFE method at 60 °C. Moreover, for all the samples, the extraction yields significantly increased by increasing the SFE temperature from 40 °C to 60 °C.

The selection of SFE processing parameters was based on the previous experience of extraction with other seeds [22,28]. These findings were in agreement with other previous studies using the SFE system at fixed pressure and different temperature from other seeds, such as *Nigells satvia* seeds [28,29] and *Swietenia mahagoni* seeds [30].

Data on the SFE extraction of OFI seed oil were limited in the literature [16,23]. The SFE extraction of Tunisian OFI oil at pressures of 180 bar, temperatures of 40 °C and extraction time of 135 min with a CO_2_ flow rate of 15 mL·s^−1^ was reported to provide a yield of 3.4% and 1.94% for spiny and thornless varieties, respectively [23]. Both results were lower than those of the present study at the same temperature (40 °C). One reason could be the difference in the pressure (180 vs. 300 bar) or the sample variability. The SFE extraction of oil from OD seeds has been reported only by Liu et al. [9]. They found that both the temperature and pressure of the SFE process affected the extraction yield. The yield first increased, reached a maximum value of around 45 °C, and then decreased at higher temperatures, while it increased more linearly by increasing the pressure. They obtained a maximum yield of 6.65%, despite the use of higher pressure with respect to the present study.

Several factors were reported to affect the oil yield, including extraction methods, geographical origin and the harvest period of the samples, fruit ripeness, and type of solvents [6,23]. In this study, the oil yield obtained by SFE was lower than the one obtained by Soxhlet extraction due to the polarity of CO_2_, a solvent suitable for the recovery of non-polar lipids. On the other side, n-hexane showed higher extractability extracting some other polar lipids and compounds in addition to the non-polar lipids. The results of the present study were in agreement with previous findings. In this context, SFE was applied to OFI from two Tunisian prickly pear seed types—the spiny (wild) and thornless (cultivated). The yielded result was significantly higher (10.32% (wild) and 8.91%) for Soxhlet compared to SFE (3.4% (spiny) and 1.94% (thornless)) [26]. In another study, the extraction of OFI using SFE was reported with a yield of 6.5% [30,31,32,33].

### 3.2. Chemical Characterization of Seed Oils

#### 3.2.1. Fatty Acid Composition

The fatty acid (FA) composition of OFI (YV1 from Yemen and I/S from Sicily) and OD (YV2) from Yemen) seed oils extracted using SFE at 40 °C, 60 °C, and Soxhlet are reported in Table 1, while the typical GC chromatographic profile is depicted in Appendix A, in which only one chromatogram is shown since no qualitative differences were observed between the different samples. The results indicated that OFI and OD seed oils were characterized by a high level of unsaturated FAs (around 83%). They were mainly composed of linoleic acid (C18:2w6), which accounted for a percentage higher than 62% in all the samples. In the present study, oleic acid (C18:1w9) content ranged from a minimum of 9.88% in YV2 extracted by SFE at 60 °C to a maximum of 13.97% in YV1 extracted by Soxhlet, immediately followed by YV1 seed oil extracted by SFE at 60 °C and the I/S sample extracted by SFE at both tested temperatures. Such results clearly pinpointed the FA composition of both *Opuntia* spp. OFI and OD seed oils were not particularly affected by the extraction method, while it was most affected by the botanical species of the investigated seeds.

The main saturated fatty acids were palmitic and stearic acids in the three samples. Palmitic acid (C16:0) ranged from a minimum of 11.8% in YV2 and YV1 obtained by Soxhlet and SFE at lower temperatures, respectively, to a maximum of 12.5–12.6% in I/S and YV1 (both from OFI seeds) obtained by SFE at 40 °C and 60 °C, respectively. As for minor FAs, the omega-3 α-linolenic acid (C18:3w3) was quantified in a higher amount in the SFE extracts of YV2 seeds, immediately followed by the SFE extracts of YV1 and I/S seeds at 60 °C and 40 °C, respectively, highlighting a good capability of supercritical CO_2_ to extract more polar FAs, even if present at low levels.

Generally, the FA quali-quantitative profile of both OFI and OD seed oils determined in the present study was in accordance with the previous literature [3,31,32], which reported linoleic acid to be the most abundant compound in the range 57.54–66.57%, followed by oleic acid in the range 15.2–24.3%, thus representing almost 80% of the total FAs and contributing to a favorable ratio UFA/SFA > than four.

#### 3.2.2. HPLC/MS Analysis of Intact Lipids

Conversely, from the total FA composition, TAGs, which represent the native lipid composition of the seeds, were rarely investigated in OFI seed oils, while no studies are reported in the literature about the TAGs composition of OD samples. Indeed, the intact lipid composition could also be affected by the extraction method since the way in which single FAs are combined in the more complex TAG structure significantly affects the polarity of the molecule compared to their affinity to supercritical CO_2_.

Table 2 reports the list of the identified compounds, while the typical LC chromatographic profile is depicted in Appendix A, in which only one chromatogram is shown since no qualitative differences were observed between different samples. The profile of the sample was reported in comparison with one of the reference standard mixtures used for LRI calculation, according to the equation reported in the insert of the figure. It is quite clear that TAGs are eluted according to their increasing hydrophobicity, expressed as partition number (PN=CN-2DB, where CN is the carbon number and DB is the number of double bonds of the FAs bound to the glycerol backbone). The use of the LRI identification strategy, in combination with the search into the homemade MS spectral library, allowed the reliable identification of 30 TAGs, many of them identified for the first time in *Opuntia* spp. seed oils.

According to the results of the ANOVA test reported in Table 2, major quantitative differences between the samples were obtained for diacylglycerols (DAGs) and Ln-containing TAGs (*p* < 0.01), apart from significant differences observed for some Linoleic (L)-containing TAGs, such as trilinolein (LLL), oleyl-dilinolein (OLL) and dioleyl-linolein (OOL), the latter partially coeluted with gondoyl-dilinolein (GLL).

Particularly, DAGs were detected at a significantly higher percentage in the YV2 sample extracted by SFE at 60 °C, while Ln-containing TAGs were quantified at higher levels in the Sicilian sample (I/S) obtained through SFE at 60 °C. Such results appeared in contrast with GC results of the FA composition, which reported the highest content of Ln in YV2 extracted by SFE. However, the GC analysis provided results regarding the total FA composition, including free FAs and monoacylglycerols, not detected through the employed LC method. Then, it could be supposed that both YV2 and I/S seed oils had a satisfactory content of Ln, extracted in major amounts through SFE compared to Soxhlet, and independently from the native form in which it was present in the seeds (TAGs, free FA or MAG).

Generally, the quali-quantitative TAGs profile of the seed oils here investigated was similar to the composition previously reported for OFI seed oil originating from Tunisia and extracted by Soxhlet using hexane, with LLL and OLL as the main components with an average percentage of 25.6% and 21.5, respectively [33]. The analyses of the present study showed a higher content of LLL, near or higher than 30% in all the samples, while OLL was contained at a lower percentage, even less than 20%, in some of the investigated samples. Such difference in the TAG composition reflects the lower content of oleic acid found from the GC analyses. On the other hand, a higher content of dilinoleyl-palmitin (LLP) was obtained, comparable to OLL (17.62–19.01%). Additionally, many Ln-containing TAGs were identified at percentages of around 0.5%, as well as other minor TAGs, not present in the work by Mannoubi et al. [33], which reported a remarkably higher content of another oleic acid (O)- containing TAG, namely OOL (11.40 vs. 3.94–6.62% of the present study).

#### 3.2.3. LC-FLD Analysis of Vitamin E

The LC-FLD analyses of vitamin E revealed the presence of γ-tocopherol as the only vitamer contained in all the OFI and OD seed oils. Such a result is in accordance with a previous work in the literature on Moroccan OFI and OD seed oils extracted using hexane, in which only γ-tocopherol was detected [34]. Other authors reported the presence of α-tocopherol and δ-tocopherol with a content lower than 2 and 10 mg/100, respectively [5,33,35]. However, γ-tocopherol was still the main vitamer, with a content ranging from 67.5 to 91.7 mg/100 g.

As shown in Table 3, γ-tocopherol ranged between 73 and 178 mg/100 g. More in detail, the Yemen OD seed oils (sample YV2) showed only a moderate variation in the γ-tocopherol content based on the extraction procedure. Conversely, the Yemen OFI seed oil (sample YV1) and the Sicilian I/S sample showed a significantly higher content (*p* < 0.01) of γ-tocopherol in the SFE extract at 60 °C. Generally, for both YV1 and I/S seed oils, the tocopherol content was affected by the extraction method, as was reported previously by Regalado-Rentería et al. [36], who compared cold pressing and maceration extraction for different *Opuntia* species. They found that the γ-tocopherol content was significantly higher in cold-pressed oils than those extracted with solvents. Additionally, the vitamin E content was highly dependent on the botanical species since YV2 seed oils contained a significantly lower content (*p* < 0.01) of γ-tocopherol than YV1. This was in agreement with previous findings [34] in which OFI seeds were richer in vitamin E than OD seeds.

#### 3.2.4. Phenolic Compounds

The extracts from the oils were analyzed by LC-PDA/ESI-MS in order to obtain a chromatographic profile to correlate with the results of the antioxidant activity. Moreover, the identification of single phenolic compounds resulted in an added value to the present work as it allowed us to identify the molecules responsible for specific biological activities. Within this context, most works in the literature reported mainly the total phenolic and/or flavonoid content, while few data were published about the detailed phenolic composition of these samples. Among them, the results of the present study were not far from those previously reported [37]. In a published study, the phenolic composition of OFI cold-pressed oils from six different locations in Morocco was investigated. Seven phenolic compounds were identified, belonging to three families, the hydroxyl cinnamic acid derivatives (p-coumaric acid, p-coumaric acid ethyl ester, ferulic acid), the hydroxyl cinnamaldehyde derivatives (ferulaldehyde) and the hydroxybenzaldehyde derivatives (4-hydroxybenzaldehyde, vanillin, syringaldehyde) with vanillin, syringaldehyde, and ferulaldehyde being the most abundant ones.

In the present study, a total of 14 phenols were identified with quali-quantitative differences among different samples and/or extraction methods. They belong to the same chemical classes reported in the literature [37] with the addition of two simple phenols, namely tyrosol and hydroxytyrosol, which are characterized by a well-known high antioxidant power [38], three benzoic acid derivatives, namely syringic acid, vanillic and homovanillic acids, and one coumarin derivative that is the 4-hydroxycoumarin. As for the benzoic acids, they could derive from vanillin oxidation (with vanillin confirmed as the most representative phenol in both prickly pear seed oils) or could be natural flavoring agents with antimicrobial properties [39]. Among cinnamic acid derivatives, p-coumaric acid and its ethyl ester were not detected, while a moderate content of cinnamic acid was observed in most of the samples under investigation. Its presence was quite conceivable since it is a central intermediate in the phenylpropanoid biosynthetic pathway, leading to the synthesis of other hydroxycinnamic derivatives, flavonoids, isoflavones, lignin, and coumarins (e.g., 4-hydroxycoumarin) [40]. Finally, among cinnamaldehyde derivatives, synapaldehyde was also identified for the first time in both OFI and OD seed oils. It is also involved in the biosynthesis of lignin [41,42], then it could be reasonably contained in the seed oil. Indeed, published studies [43] detected glucoside derivatives of both synapoyl and feruloyl acids in the ethanolic extracts of the seeds, thus supporting the tentative identification of the present study. All these findings are summarized in Appendix A, which lists the tentatively identified phenols, their chemical family, and their percentage in the analyzed oils.

To the best of the authors‘ knowledge, there were no previous studies on the detailed profiling of phenolic compounds in OD seed oil. Actually, the YV2 sample showed the richest phenolic profile, especially when Soxhlet extraction was used, followed by SFE at 60 °C. The HPLC-PDA chromatograms of YV2 seed oils extracted through the three methods are shown in Figure 2D–F, which point out the different signal intensities of the samples obtained through Soxhlet (intensity scale multiplied by five compared to the SFE). Soxhlet extraction provided the most intense signal of phenolic compounds in the case of the Sicilian sample (I/S) (Figure 2G–I highlighted an intensity scale multiplied by 4 compared to the corresponding SFE). Strangely, similar profiles were obtained for the YV1 samples (Figure 2A–C), independently from the extraction procedure. As for the quali-quantitative composition, the SFE extracts were dominated by the vanillin content, which decreased in the Soxhlet extract due to the simultaneous extraction of a major number of compounds (e.g., vanillin isomer, syringic, ferulic, and cinnamic acids) with a higher overall yield (details on the quali-quantitative composition of phenolic compounds in OFI and OD seed oil was shown in Appendix A).

### 3.3. Antioxidant Activity of OFI and OD Seeds Oils

The total antioxidant activity of YV1, YV2, and I/S extracted oils using SFE at different temperatures, and Soxhlet was assessed spectrophotometrically using the DPPH assay.

The antioxidant activity was equal to 47.78 ± 0.41, 34.96 ± 5.323, and 42.16 ± 1.452 mg Trolox eq./g oil in YV1 oil samples extracted by Soxhlet, SFE at 40 °C and 60 °C, respectively (Figure 3). Conversely, YV2 showed significant differences in the antioxidant activity depending on the extraction with 52.86 ± 1.526, 30.21 ± 2.78, and 42.67 ± 0.1718 mg Trolox eq./g oil for the Soxhlet, SFE at 40 and 60 °C samples, respectively.

The Soxhlet extraction of the I/S sample provided the highest antioxidant activity (61.54 ± 1.8) compared to all other samples, despite no significant differences (*p* > 0.01) observed with respect to YV2 oil extracted by Soxhlet. The differences in antioxidant activity among different samples extracted by Soxhlet might be ascribed to the geographical location where the seeds were harvested. Liu et al. [44] also reported that the geographical location significantly affected the phenolic composition and antioxidant activities of the extracts.

For the antioxidant activity, differences between the three extraction methods were observed, with Soxhlet appearing more efficient than SFE at 60 °C, which was more efficient than SFE at 40 °C for both YV2 and I/S samples, while similar profiles were obtained for YV1 with all the extraction procedures.

The results of the present study on the antioxidant activity of OFI seeds were comparable to previous findings, in which another extraction method, i.e., cold pressing, was used for the production of Algerian seed oil [45]. It was reported that Soxhlet extraction provided seed oils with higher antioxidant activity, as also shown in other studies if the extraction solvent was carefully selected [5]. For instance, the results of the free radical scavenging activity of OFI originating from Yemen and extracted by different solvents, namely n-hexane, petroleum ether, and chloroform-methanol (2:1, *v*/*v*) showed that the oil extracted using chloroform-methanol (2:1, *v*/*v*), exhibited the highest antioxidant activity (87%) towards the DPPH radical, especially compared to petroleum ether (76%) [5]. Another work on the determination of phenolic compounds and antioxidant activity of Algerian OFI seed oil indicated that the best results of the antioxidant capacity of seeds were obtained when 75% of acetone (rather than ethanol and methanol) were mixed with water and used as an extraction solvent, reaching an antioxidant capacity ranging from 50% to 95% by increasing the extraction temperature [46]. As for the oil extracted from OD seeds, a previous work highlighted a notable antioxidant activity of the seed oil obtained using SFE at 45 °C, 46.96 MPa, and 2.79 h of extraction time. The radical scavenging activity was nearly comparable to a reference ascorbic acid solution [9].

## 4. Conclusions

SFE and Soxhlet extraction was applied for the recovery of oils from OFI and OD seeds. Higher oil yields were obtained by Soxhlet extraction compared to SFE. Fatty acid profiles were investigated. Linoleic acid was the major compound, followed by oleic acid and palmitic acid for both SFE and Soxhlet extraction methods. SFE allowed for better extraction of linolenic acid compared to Soxhlet extraction. A total of 30 TAGs were identified in both OFI and OD seed oils for the first time. The main TAGs components were the same for the SFE and Soxhlet extraction. LLL was the major TAG, followed by OLL and SLL+OLP. The phenolic profiles showed 14 different compounds, with vanillin, 4-hydroxy benzaldehyde, vanillic acid, and hydroxytyrosol appearing as the most abundant phenols. The best results for OFI and OD SFE extracted oils were obtained at 60 °C, which provided a higher oil yield, antioxidant activity, linolenic acid, and γ-tocopherol contents compared to the oils extracted at 40 °C. In contrast, Soxhlet allowed for better extraction of total phenolic compounds with higher antioxidant activity. The results clearly highlight that an in-deep study on the optimization of SFE processing conditions is still needed to obtain oils with a higher yield, phenolic profile, and high antioxidant activity. This would allow the application of SFE as a green extraction technology to obtain oils that are useful for the formulation of functional food or nutraceutical products.

## Figures and Tables

**Figure 1 foods-12-00618-f001:**
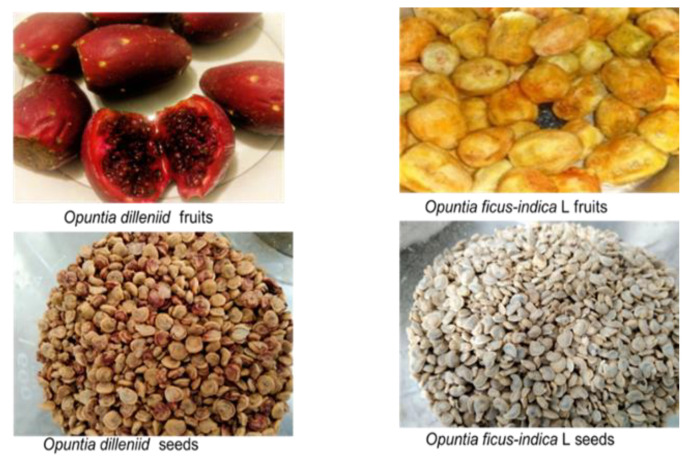
Appearance of *Opuntia ficus-indica* L. and *Opuntia dillenii* fruits and seeds.

**Figure 2 foods-12-00618-f002:**
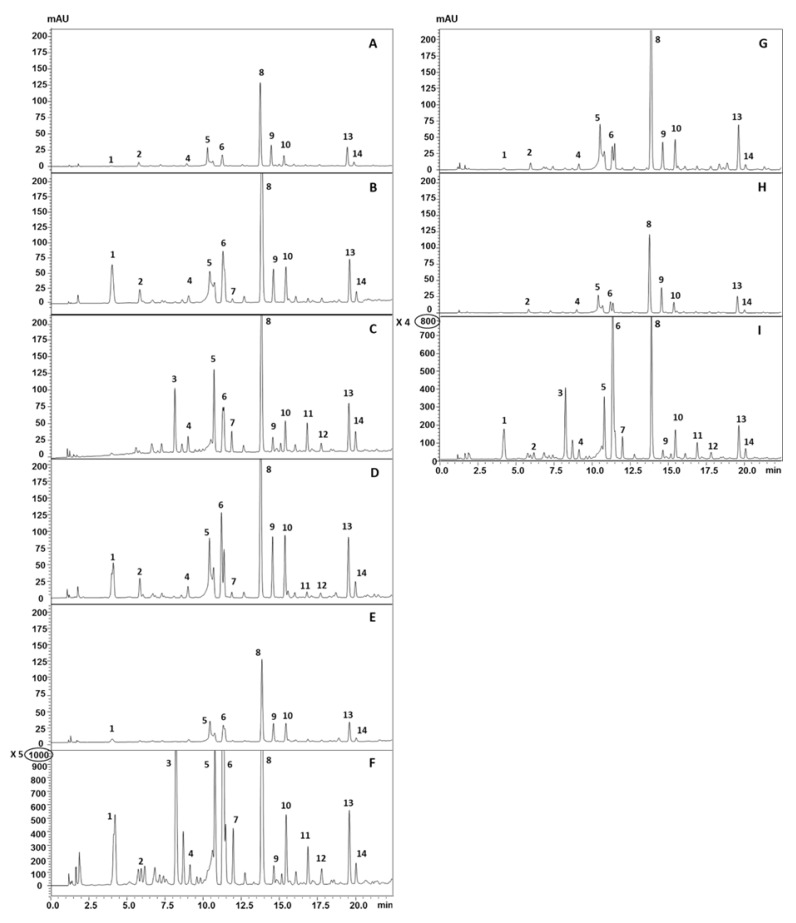
HPLC-PDA chromatograms of phenolic compounds in YV1 (**A**–**C**), YV2 (**D**–**F**), and I/S (**G**–**I**) seed oils extracted using SFE at 60 °C (**A**,**D**,**G**), 40 °C (**B**,**E**,**H**), and Soxhlet extraction (**C**,**F**,**I**).

**Figure 3 foods-12-00618-f003:**
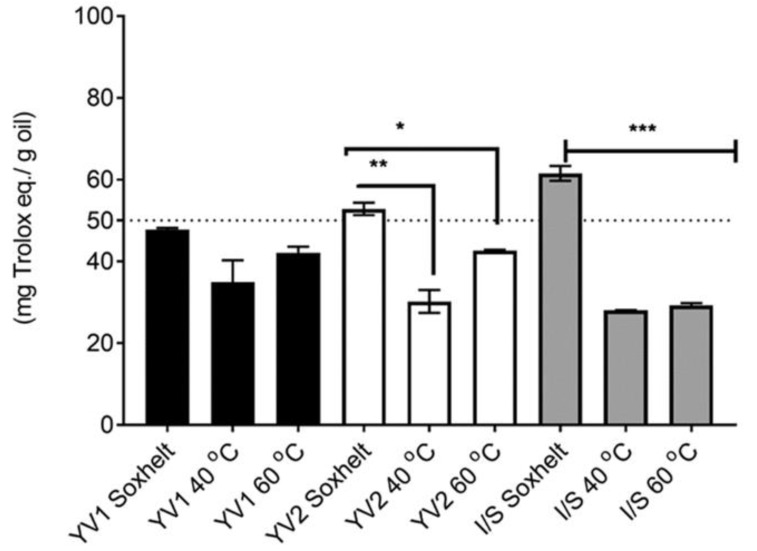
Antioxidant activity (mg Trolox eq./g oil) of OFI and OD oils extracted by SFE at 40 °C, 60 °C, and Soxhlet. Among each variety, different symbols (* *p* < 0.05, ** *p* < 0.01, *** *p* < 0.001) indicate significant differences compared to Soxhlet extraction.

**Table 1 foods-12-00618-t001:** Fatty acid (FA) composition of YV1, YV2, and I/S seed oils extracted using SFE and Soxhlet methods.

FA	YV1 (Area%)			YV2 (Area%)			I/S (Area%)		
	SFE 40 °C	SFE 60 °C	Soxhlet	SFE 40 °C	SFE 60 °C	Soxhlet	SFE 40 °C	SFE 60 °C	Soxhlet
C14:0	0.08 ± 0.00 ^b^	0.09 ± 0.00 ^ab^	0.09 ± 0.01 ^ab^	0.09 ± 0.01 ^a^	0.09 ± 0.01 ^a^	0.09 ± 0.01 ^ab^	0.09 ± 0.01 ^a^	0.09 ± 0.01 ^a^	0.09 ± 0.01 ^ab^
C16:0	11.81 ± 0.01 ^d^	12.62 ± 0.10 ^a^	12.35 ± 0.01 ^b^	11.97 ± 0.01 ^c^	12.06 ± 0.01 ^c^	11.76 ± 0.01 ^d^	12.52 ± 0.02 ^a^	12.39 ± 0.02 ^b^	11.86 ± 0.01 ^d^
C16:1w7	0.67 ± 0.01 ^e^	0.74 ± 0.00 ^a^	0.74 ± 0.01 ^a^	0.7 ± 0.01 ^bcd^	0.72 ± 0.01 ^ab^	0.69 ± 0.01 ^cde^	0.71 ± 0.01 ^abc^	0.68 ± 0.01 ^de^	0.7 ± 0.01 ^bcd^
C16:1w5	0.05 ±0.01 ^c^	0.07 ± 0.01 ^b^	0.08 ± 0.01 ^ab^	0.04 ± 0.01 ^cd^	0.03 ± 0.01 ^e^	0.03 ± 0.01 ^de^	0.08 ± 0.01 ^ab^	0.09 ± 0.01 ^a^	0.04 ± 0.01 ^cde^
C16:2w4	0.08 ± 0.01 ^c^	0.07 ± 0.01 ^e^	0.07 ± 0.01 ^cde^	0.11 ± 0.01 ^b^	0.13 ± 0.01 ^a^	0.13 ± 0.01 ^a^	0.08 ± 0.01 ^cd^	0.07 ± 0.00 ^de^	0.11 ± 0.01 ^b^
C17:0	0.04 ± 0.01 ^bc^	0.04 ± 0.01 ^c^	0.04 ± 0.01 ^bc^	0.05 ± 0.01 ^abc^	0.05 ± 0.01 ^a^	0.05 ± 0.01 ^a^	0.05 ± 0.01 ^ab^	0.04 ± 0.01 ^c^	0.05 ± 0.02 ^ab^
C18:0	3.1 ± 0.01 ^e^	2.69 ± 0.01 ^h^	2.7 ± 0.01 ^h^	3.43 ± 0.01 ^d^	3.88 ± 0.01 ^b^	3.91 ± 0.01 ^a^	2.75 ± 0.01 ^g^	2.92 ± 0.01 ^f^	3.69 ± 0.01 ^c^
C18:1w9	11.7 ± 0.01 ^d^	13.8 ± 0.01 ^b^	13.97 ± 0.01 ^a^	11.22 ± 0.01 ^e^	9.88 ± 0.01 ^h^	10.02 ± 0.01 ^g^	13.65 ± 0.01 ^c^	13.82 ± 0.01 ^b^	10.76 ± 0.01 ^f^
C18:1w7	4.82 ± 0.01 ^f^	5.41 ± 0.01 ^b^	5.49 ± 0.01 ^a^	4.82 ± 0.02 ^f^	4.65 ± 0.01 ^g^	4.83 ± 0.01 ^f^	5.16 ± 0.01 ^d^	5.2 ± 0.01 ^c^	4.97 ± 0.01 ^e^
C18:1w3	0.11 ± 0.01 ^b^	0.16 ± 0.01 ^a^	0.19 ± 0.01 ^a^	0.1± 0.01 ^bc^	0.06 ± 0.01 ^c^	0.08 ± 0.01 ^bc^	0.2 ± 0.01 ^a^	0.19 ± 0.02 ^a^	0.1 ± 0.01 ^bc^
C18:2w6	66.09 ± 0.01 ^b^	62.78 ± 0.04 ^e^	62.7 ± 0.03 ^e^	65.79 ± 0.02 ^c^	66.59 ± 0.01 ^a^	66.52 ± 0.02 ^a^	63.12 ± 0.03 ^d^	62.77 ± 0.03 ^e^	65.83 ± 0.02 ^c^
C18:3w3	0.23 ± 0.01 ^e^	0.37 ± 0.01 ^b^	0.28 ± 0.01 ^d^	0.42 ± 0.01 ^a^	0.42 ± 0.01 ^a^	0.34 ± 0.00 ^c^	0.39 ± 0.01 ^b^	0.34 ± 0.01 ^c^	0.32 ± 0.01 ^c^
C20:0	0.33 ± 0.01 ^d^	0.27 ± 0.01 ^f^	0.3 ± 0.01 ^e^	0.34 ± 0.01 ^c^	0.4 ± 0.01 ^ab^	0.41 ± 0.01 ^a^	0.29 ± 0.01 ^e^	0.34 ± 0.02 ^cd^	0.38 ± 0.02 ^b^
C20:1w9	0.16 ± 0.01 ^b^	0.15 ± 0.01 ^b^	0.16 ± 0.01 ^b^	0.16 ± 0.01 ^b^	0.19 ± 0.01 ^a^	0.21 ± 0.01 ^a^	0.14 ± 0.01 ^b^	0.14 ± 0.01 ^b^	0.2 ± 0.01 ^a^
C20:1w7	0.13 ± 0.01 ^d^	0.17 ± 0.01 ^bc^	0.18 ± 0.01 ^ab^	0.14 ± 0.01 ^cd^	0.12 ± 0.01 ^d^	0.13 ± 0.01 ^d^	0.17 ± 0.01 ^ab^	0.19 ± 0.01 ^a^	0.14 ± 0.01 ^d^
C22:0	0.22 ± 0.01 ^cd^	0.16 ± 0.01 ^g^	0.19 ± 0.01 ^ef^	0.2± 0.01 ^de^	0.27 ± 0.01 ^b^	0.29 ± 0.01 ^a^	0.18 ± 0.01 ^f^	0.22 ± 0.01 ^c^	0.27 ± 0.01 ^b^
C22:1w7	0.25 ± 0.01 ^de^	0.23 ± 0.01 ^e^	0.27 ± 0.01 ^c^	0.25 ± 0.01 ^d^	0.27 ± 0.01 ^bc^	0.3 ± 0.01 ^a^	0.25 ± 0.01 ^d^	0.29 ± 0.01 ^ab^	0.29 ± 0.02 ^a^
C24:1w9	0.06 ± 0.01 ^bc^	0.07 ± 0.01 ^ab^	0.06 ± 0.01 ^ab^	0.05 ± 0.01 ^c^	0.06 ± 0.01 ^bc^	0.06 ± 0.01 ^ab^	0.06 ± 0.01 ^bc^	0.07 ± 0.02 ^a^	0.06 ± 0.01 ^ab^
SFA	15.67 ± 0.38 ^f^	15.97 ± 0.29 ^d^	15.8 ± 0.22 ^e^	16.15 ± 0.42 ^c^	16.87 ± 0.11 ^a^	16.66 ± 0.21 ^b^	15.99 ± 0.11 ^d^	16.15 ± 0.22 ^c^	12.79 ± 0.11 ^g^
MUFA	17.93 ± 0.97 ^d^	20.74 ± 0.95 ^b^	21.14 ± 0.83 ^a^	17.49 ± 0.64 ^e^	16.00 ± 0.62 ^h^	16.32 ± 0.71 ^g^	20.42 ± 0.82 ^c^	20.67 ± 0.72 ^b^	17.26 ± 0.82 ^f^
PUFA	66.4± 0.91 ^c^	63.22 ± 0.94 ^f^	63.06 ± 0.93 ^g^	66.32 ± 0.93 ^cd^	67.13 ± 0.91 ^a^	66.99 ± 0.81 ^b^	63.59 ± 0.83 ^e^	63.18 ± 0.73 ^f^	66.26 ± 0.82 ^d^
PUFA/SFA	4.24 ± 0.02^b^	3.96 ± 0.02 ^f^	3.99 ± 0.01 ^e^	4.11 ± 0.02 ^c^	3.98 ± 0.01 ^ef^	4.02 ± 0.01 ^d^	3.98 ± 0.01 ^ef^	3.91 ± 0.01 ^g^	5.18 ± 0.04 ^a^

Values are presented as means ± SD (*n* = 3). Different letters within a row indicate significant differences, a–h at *p* < 0.01.

**Table 2 foods-12-00618-t002:** TAG and DAG composition of composition of YV1, YV2, and I/S seeds oils extracted using SFE and Soxhlet.

Compound	YV1 (Area%)			YV2 (Area%)			I/S (Area%)		
	SFE 40 °C	SFE 60 °C	Soxhlet	SFE 40 °C	SFE 60 °C	Soxhlet	SFE 40 °C	SFE 60 °C	Soxhlet
LL	1.44 ± 0.20 ^c^	1.30 ± 0.10 ^c^	1.03 ± 0.47 ^c^	2.99 ± 0.14 ^b^	3.62 ± 0.25 ^a^	2.70 ± 0.20 ^b^	1.03 ± 0.12 ^c^	0.95 ± 0.15 ^c^	2.51 ± 0.12 ^b^
OL+LP	1.18 ± 0.10 ^c^	1.02 ± 0.10 ^cd^	0.89 ± 0.40 ^cd^	1.91 ± 0.16 ^ab^	2.21 ± 0.12 ^a^	1.66 ± 0.07 ^b^	0.74 ± 0.12 ^d^	0.88 ± 0.06 ^cd^	1.60 ± 0.16 ^b^
LnLnLn	0.10 ± 0.02 ^b^	0.14 ± 0.02 ^b^	0.10 ± 0.04 ^b^	0.56 ± 0.24 ^b^	0.27 ± 0.04 ^b^	0.25 ± 0.03 ^b^	0.51 ± 0.08 ^b^	1.23 ± 0.19 ^a^	0.45 ± 0.07 ^b^
LLnLn	0.49 ± 0.07 ^e^	0.51 ± 0.12 ^de^	0.25 ± 0.07 ^e^	1.60 ± 0.32 ^ab^	1.02 ± 0.11 ^cd^	0.60 ± 0.06 ^de^	1.33 ± 0.16 ^bc^	2.07 ± 0.19 ^a^	0.41 ± 0.03 ^e^
LnLnP	0.18 ± 0.03 ^bcd^	0.15 ± 0.02 ^cd^	0.14 ± 0.05 ^d^	0.40 ± 0.34 ^ab^	0.36 ± 0.07 ^abc^	0.44 ± 0.09 ^a^	0.30 ± 0.06 ^abcd^	0.42 ± 0.04 ^a^	0.34 ± 0.08 ^abc^
OLnLn	0.57 ± 0.10 ^c^	0.61 ± 0.04 ^c^	0.62 ± 0.02 ^c^	1.02 ± 0.09 ^b^	0.69 ± 0.04 ^c^	0.60 ± 0.02 ^c^	1.04 ± 0.09 ^b^	1.33 ± 0.04 ^a^	0.56 ± 0.07 ^c^
LLL	34.08 ± 0.41a ^bcd^	31.99 ± 0.68 ^bcd^	31.63 ± 0.98 ^cd^	32.72 ± 1.10 ^abcd^	34.34 ± 0.70 ^abc^	35.24 ± 0.56 ^a^	31.08 ± 0.61 ^de^	29.44 ± 0.92 ^e^	35.00 ± 1.56 ^abc^
LnPPo+OLLn	0.38 ± 0.07 ^c^	0.28 ± 0.10 ^c^	0.46 ± 0.14b ^c^	0.31 ± 0.01 ^c^	0.49 ± 0.03 ^cb^	0.78 ± 0.07 ^ab^	0.46 ± 0.18 ^bc^	0.88 ± 0.06 ^a^	0.52 ± 0.08 ^abc^
OLL	19.36 ± 0.54 ^bcd^	21.28 ± 0.10 ^ab^	22.10 ± 0.88 ^a^	18.20 ± 0.19 ^cde^	16.57 ± 0.99 ^e^	17.49 ± 0.38 ^de^	21.17 ± 0.15 ^ab^	19.82 ± 0.26 ^bc^	18.47 ± 0.34 ^cde^
LLP	18.80 ± 0.57	18.70 ± 0.23	17.80 ± 1.08	18.25 ± 0.46	18.77 ± 0.08	18.45 ± 1.02	19.01 ± 1.31	18.22 ± 1.11	17.62 ± 0.57
OOL+GLL	5.29 ± 0.20 ^bc^	6.19 ± 0.10 ^a^	6.62 ± 0.32 ^a^	4.63 ± 0.12 ^cd^	3.94 ± 0.17 ^d^	4.19 ± 0.06 ^d^	6.05 ± 0.17 ^ab^	6.05 ± 0.30 ^ab^	4.51 ± 0.29 ^cd^
SLL+OLP	10.18 ± 0.12	9.44 ± 0.20	9.84 ± 0.35	9.46 ± 0.42	9.59 ± 0.60	10.10 ± 0.34	9.35 ± 0.26	9.85 ± 0.74	9.70 ± 0.25
LPP	1.49 ± 0.11	1.67 ± 0.09	1.67 ± 0.19	1.46 ± 0.12	1.37 ± 0.10	1.30 ± 0.12	1.59 ± 0.31	1.75 ± 0.23	1.17 ± 0.14
GOL+C22:1LL	0.57 ± 0.07	0.57 ± 0.01	0.67 ± 0.089	0.68 ± 0.13	0.49 ± 0.05	0.66 ± 0.11	0.58 ± 0.08	0.71 ± 0.07	0.71 ± 0.17
SOL+OOP	3.45 ± 0.23 ^ab^	3.71 ± 0.11 ^a^	3.61 ± 0.23 ^a^	2.89 ± 0.15 ^b^	2.89 ± 0.23 ^b^	3.06 ± 0.22 ^ab^	3.38 ± 0.27 ^ab^	3.52 ± 0.18 ^ab^	2.91 ± 0.04 ^b^
SLP	1.04 ± 0.08	1.15 ± 0.14	1.14 ± 0.21	1.25 ± 0.15	1.24 ± 0.19	1.11 ± 0.16	1.00 ± 0.07	1.377 ± 0.16	0.97 ± 0.11
C22:1OL+C24:1LL	0.29 ± 0.02	0.25 ± 0.06	0.21 ± 0.04	0.33 ± 0.03	0.17 ± 0.01	0.25 ± 0.09	0.29 ± 0.13	0.31 ± 0.01	0.17 ± 0.02
C22:1LP	0.13 ± 0.40 ^ab^	0.12 ± 0.01 ^ab^	0.11 ± 0.10 ^ab^	0.06 ± 0.01 ^b^	0.11 ± 0.02 ^ab^	0.17 ± 0.01 ^a^	0.14 ± 0.04 ^ab^	0.15 ± 0.05 ^ab^	0.21 ± 0.02
AOL+BLL	0.60 ± 0.30	0.55 ± 0.16	0.53 ± 0.05	0.63 ± 0.10	0.79 ± 0.23	0.69 ± 0.11	0.60 ± 0.15	0.69 ± 0.19	0.63 ± 0.07
SOO+C20:1OP+SSL	0.40 ± 0.30	0.37 ± 0.05	0.32 ± 0.045	0.44 ± 0.14	0.40 ± 0.16	0.43 ± 0.09	0.37 ± 0.02	0.39 ± 0.03	0.45 ± 0.13

Mean values with different letters are significantly different, a–e at *p* < 0.01. Fatty acid legend: L = linoleic acid C18:2n6; O = oleic acid C18:1n9; P = palmitic acid C16:0; Ln = α-linolenic acid C18:3n3; Po = palmitoleic acid C16:1n7; S = stearic acid C18:0; A = arachidic acid C20:0; B = behenic acid C22.

**Table 3 foods-12-00618-t003:** Vitamin E (γ-tocopherol) content in YV1, YV2, and I/S seeds oils extracted using SFE and Soxhlet.

	YV1			YV2			I/S		
Vitamin E (mg/100 g)	SFE 40 °C 108.1 ± 0.82 ^c^	SFE 60 °C 177.5 ± 0.23 ^a^	Soxhlet 135.40 ± 0.55 ^b^	SFE 40 °C 81.6 ± 0.16 ^g^	SFE 60 °C 89.4 ± 0.47 ^f^	Soxhlet 88.9 ± 1.32 ^e^	SFE 40 °C 73.0 ± 0.27 ^h^	SFE 60 °C 105.2 ± 0.51 ^d^	Soxhlet 89.7 ± 0.07 ^i^

Mean values with different letters are significantly different, a–i at *p* < 0.01.

## Data Availability

Data are contained within the article or Appendix A.

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
