# Peer review of "Supercritical Fluid Extraction of Oils from Cactus *Opuntia ficus-indica* L. and *Opuntia dillenii* Seeds"

_foods, 2023, doi:10.3390/foods12030618_

Round 1
Reviewer 1 Report
The manuscript entiled, “Supercritical fluid extraction of Cactus Opuntia ficus-indica L and Opuntia dillenii seeds oil: antioxidant activity, chemical characterization and comparison with a conventional solvent extraction” describes the extraction of cactus Opuntia ficus-indica L and Opuntia dillenii seeds oil using Supercritical fluid extraction and their biological characteristics. I think this manuscript needs a lot of correction before considering for Foods. However, some concerns must be clarified.
- I think the abstract's representation is too general; the authors need to add important results to make it more relevant.
- The title of the manuscript should be modified. Reduce the number of words and be concise.
- It is unclear why the authors chose 300 bar pressure and 40 °C, and 60 °C for oil extraction. What was the proximate composition of dried seeds? What was the average diameter of the sample particle used for oil extraction? The particle size of the samples directly affects the oil extraction during supercritical fluid extraction.
- Line number 66, cold press and solvent extraction frailties, need to add some reference.
- In line 113, the authors need to add a full stop and author name.
- Line 171, 181, 207, the authors should give the name of the researcher; X et al. (26). In any scientific paper, in the sentence, the author should provide the researcher's name in the whole manuscript.
- Section 3.1, line 226, it seems that, Soxhlet extraction yield was high compared to SFE. Why need to use SFE? What is the reason for lower yield by SFE?
- Soxhlet extraction of I/S sample provided the highest antioxidant activity (61.54 ± 1.8) compared to all other samples. What is the reason behind it? Please explain.
- The authors should provide clear Figures 3 & 4; it is tough for readers to understand in this format.
- In the introduction, the authors should clarify the purpose of the work more precisely. I think authors should check the whole manuscript critically to find their grammatical mistakes.
Author Response
Response to Reviewer 1
Point 1: I think the abstract's representation is too general; the authors need to add important results to make it more relevant.
Authors’ response: The abstract was modified as suggested by the reviewer.
Point 2: The title of the manuscript should be modified. Reduce the number of words and be concise.
Authors’ response: The title was modified and shorten.
Point 3. It is unclear why the authors chose 300 bar pressure and 40 °C, and 60 °C for oil extraction. What was the proximate composition of dried seeds? What was the average diameter of the sample particle used for oil extraction? The particle size of the samples directly affects the oil extraction during supercritical fluid extraction.
Authors’ response: An explanation was added to the manuscript on the choice of the operative conditions. Information on the average diameter of the particle size of the powder were also added. The proximate composition of dried seeds was not evaluated as it is already reported in the literature:
https://pubmed.ncbi.nlm.nih.gov/9950087/
https://www.tandfonline.com/doi/abs/10.3109/09637486.2011.552569?journalCode=iijf20
Point 4. Line number 66, cold press and solvent extraction frailties, need to add some reference.
Authors’ response: References were added.
Point 5. In line 113, the authors need to add a full stop and author name.
Authors’ response: The reference was corrected.
Point 6. Line 171, 181, 207, the authors should give the name of the researcher; X et al. (26). In any scientific paper, in the sentence, the author should provide the researcher's name in the whole manuscript.
Authors’ response: The references were corrected.
Point 7. Section 3.1, line 226, it seems that, Soxhlet extraction yield was high compared to SFE. Why need to use SFE? What is the reason for lower yield by SFE?
Authors’ response: The main aim of this study was to use the SFE extraction as an alternative and green technique compared to Soxhlet for the production of oils from OFI and OD seeds. Even though, Soxhlet extraction using hexane provided a higher yield compared to SFE, hexane should be considered with care when used in producing oils for health purposes. Also, solvent extraction has environmental hazardous and negative implication for the operator safety. Moreover, they might have a threat to consumer health if the organic solvents are not completely removed. This is the reason why SFE should be preferred over conventional solvent extractions. In this study, the processing parameters for SFE extraction were not optimized as a consequence a lower yield was obtained. An explanation was added to the manuscript.
Point 8. Soxhlet extraction of I/S sample provided the highest antioxidant activity (61.54 ± 1.8) compared to all other samples. What is the reason behind it? Please explain.
Authors’ response: An explanation was added to the manuscript.
Point 8. The authors should provide clear Figures 3 & 4; it is tough for readers to understand in this format.
Authors’ response: Figures 3 and 4 were modified as suggested by the reviewer.
Point 9. In the introduction, the authors should clarify the purpose of the work more precisely. I think authors should check the whole manuscript critically to find their grammatical mistakes.
Authors’ response: The introduction was revised and two main paragraphs were added. Grammatical mistakes were checked throughout the manuscript.

Reviewer 2 Report
The present paper deals with soxhlet and SFE for extraction of oil from Cactus Opuntia ficus-indica L and Opuntia dillenii seeds. The research is very informative as authors’ highlighted use of underutilised wild crop. However there are various flaws in the manuscript that needs to be addressed. The following comments are suggested for improvement of the manuscript.
Abstract
Line 23: Write “soxhlet extraction”
Abstract is not informative. Authors should mention extraction processing parameters used for both soxhlet and SFE method. Results should be written with numerical data and statistical analysis.
Introduction
Line 42: which diseases and disorders? By virtue of which properties and constituents, Opuntia ficus-indica L (OFI) and Opuntia dillenii (OD) are used in medicine. Explain in detail.
Line 45: Again follow the same as suggested for line 42.
Introduction is very poor. Authors shall write about the chemical and nutritional composition of OFI and OD. Authors only mention about the disadvantages of soxhlet extraction and advantages of SFE but this need to be supported with research findings by other scientists on use of soxhlet and SFE.
Authors shall write the hypothesis of research.
Materials and Methods
Supercritical Fluid Extraction: Extraction was conducted with only two set of temperatures. Why the effect of pressure and CO2 flow rate were not analysed?
Statistical Analysis part is missing. How the significant difference between soxhlet and SFE for obtained data was analysed?
Results and Discussion
Why the yield in SFE was lesser than that in soxhlet extraction? Explain the reason. Also explain whether it is in accordance with findings by other researchers. Explain the statistical difference between yield from SFE and soxhlet method.
Table 1: How statistical significance was analysed? It is not explained in materials and methods.
Conclusion
In conclusion, again the summery of the paper is written. Authors should focus on analysis most suitable method between soxhlet and SFE method. Authors should conclude with recommendation of appropriate method for extraction. Also explain the practical application of this research.
Manuscript needs major grammatical revision.
Author Response
Response to Reviewer 2
Point 1. Abstract
Line 23: Write “Soxhlet extraction”
Authors’ response: Soxhlet extraction was added.
Point 2. Abstract is not informative. Authors should mention extraction processing parameters used for both Soxhlet and SFE method. Results should be written with numerical data and statistical analysis.
Authors’ response: Abstract was modified to contain important results.
Point 3: Line 42: which diseases and disorders? By virtue of which properties and constituents, Opuntia ficus-indica L (OFI) and Opuntia dillenii (OD) are used in medicine. Explain in detail.
Authors’ response: The description was deepen.
Point 4 Line 45: Again, follow the same as suggested for line 42.
Authors’ response: The sentence was deleted “OFI seed oil is also used in traditional medicine for diseases prevention.” The following paragraph present in the Introduction described in detail the medicinal and pharmacological uses of the OFI seed oil: “On the other hand, an impressive number of studies have recently shown that the seed oils of both OFI and OD showed good nutritional value and multiple health bene-fits including antioxidant activity both in vivo [10] and in vitro [11, 8], antimicrobial [12], anti-diabetic [3, 13, 14], lipid lowering effect [15], in vitro anticancer effect [16], anti-inflammatory and antiulcer effects [10, 11].”
Point 5. Introduction is very poor. Authors shall write about the chemical and nutritional composition of OFI and OD. Authors only mention about the disadvantages of soxhlet extraction and advantages of SFE but this need to be supported with research findings by other scientists on use of soxhlet and SFE.
Authors’ response: The introduction was revised as suggested by the reviewer.
Point 6. Authors shall write the hypothesis of research.
Authors’ response: The hypothesis of research together with the aim of the study was reported (lines 101-109 revised version of the manuscript).
Materials and Methods
Point 7. Supercritical Fluid Extraction: Extraction was conducted with only two set of temperatures. Why the effect of pressure and CO2 flow rate were not analyzed?
Authors’ response: In this study, the choice of operating at 60 °C was in accordance with typical SFE temperatures used for the recovery of oils from different seeds. Two temperatures were tested to see if increasing it from 40 to 60 °C an increase of the yield could be obtained. Pressure was set to the highest value that the system could stand as it is well know that increasing the pressure the yield of oils increases. CO2 flow rate was not analyzed as based on preliminary experiments no effect was detected on the oil yield.
Point 8. Statistical Analysis part is missing. How the significant difference between soxhlet and SFE for obtained data was analyzed?
Authors’ response: Statistical analysis was added to the material and method section.
Results and Discussion
Point 9. Why the yield in SFE was lesser than that in Soxhlet extraction? Explain the reason. Also explain whether it is in accordance with findings by other researchers. Explain the statistical difference between yield from SFE and Soxhlet method.
Authors’ response: A paragraph was added to the manuscript explaining why the yield of SFE was lower compared to Soxhlet. The results were compared to findings published by other researchers.
Point 10. Table 1: How statistical significance was analyzed? It is not explained in materials and methods.
Authors’ response: Statistical analysis was added to the material and method section.
Conclusion
Point 11. In conclusion, again the summery of the paper is written. Authors should focus on analysis most suitable method between Soxhlet and SFE method. Authors should conclude with recommendation of appropriate method for extraction. Also explain the practical application of this research.
Authors’ response: The conclusion was modified as suggested by the reviewer.

Round 2
Reviewer 1 Report
I would like to thank the authors for resubmitting their revised manuscript. I have checked the revised manuscript entitled, “Supercritical fluid extraction of Cactus Opuntia ficus-indica L and Opuntia dillenii seeds oil: antioxidant activity, chemical characterization and comparison with a conventional solvent extraction”. I have some minor comments before the acceptance of this manuscript.
- The response of authors against my comment (Section 3.1, line 226, it seems that, Soxhlet extraction yield was high compared to SFE. Why need to use SFE? What is the reason for lower yield by SFE?) is not satisfactory. The actual cause is, supercritical CO2 is highly suitable for extracting the non-polar lipids whereas n-hexane has higher extractability and can extract some other polar lipids and compounds with the non-polar lipids. It is the reason behind the higher extraction yield on n-hexane comparing supercritical CO2 extraction. I am suggesting authors to cite the mentioned articles (https://doi.org/10.3390/md20010070, https://doi.org/10.1111/jfpp.14892) to improve their discussion in section 3.1. of their revised article.
Author Response
Reviewer 1
I would like to thank the authors for resubmitting their revised manuscript. I have checked the revised manuscript entitled, “Supercritical fluid extraction of Cactus Opuntia ficus-indica L and Opuntia dillenii seeds oil: antioxidant activity, chemical characterization and comparison with a conventional solvent extraction”. I have some minor comments before the acceptance of this manuscript.
- The response of authors against my comment (Section 3.1, line 226, it seems that, Soxhlet extraction yield was high compared to SFE. Why need to use SFE? What is the reason for lower yield by SFE?) is not satisfactory. The actual cause is, supercritical CO2 is highly suitable for extracting the non-polar lipids whereas n-hexane has higher extractability and can extract some other polar lipids and compounds with the non-polar lipids. It is the reason behind the higher extraction yield on n-hexane comparing supercritical CO2 extraction. I am suggesting authors to cite the mentioned articles (https://doi.org/10.3390/md20010070, https://doi.org/10.1111/jfpp.14892) to improve their discussion in section 3.1. of their revised article.
Authors´ answer: The authors thank the reviewer for the comment. The discussion was improved as suggested.

Reviewer 2 Report
Thanks for revising the article.
Author Response
The authors thank the reviewer for his valuable corrections.